# Temporal and regional trends of antibiotic use in long-term aged care facilities across 39 countries, 1985-2019: Systematic review and meta-analysis

**Magdalena Z. Raban** [ORCID]*, **Peter J. Gates, Claudia Gasparini, Johanna I. Westbrook**

Centre for Health Systems and Safety Research, Australian Institute of Health Innovation, Macquarie University, Sydney, New South Wales, Australia

* magda.raban@mq.edu.au

**Data Availability Statement:** All data used in the meta-analyses are available from DRYAD (https://doi.org/10.5061/dryad.44j0zpcd9). All other data is

## Abstract

### Background

Antibiotic misuse is a key contributor to antimicrobial resistance and a concern in long-term aged care facilities (LTCFs). Our objectives were to: i) summarise key indicators of systemic antibiotic use and appropriateness of use, and ii) examine temporal and regional variations in antibiotic use, in LTCFs (PROSPERO registration CRD42018107125).

### Methods & findings

Medline and EMBASE were searched for studies published between 1990–2021 reporting antibiotic use rates in LTCFs. Random effects meta-analysis provided pooled estimates of antibiotic use rates (percentage of residents on an antibiotic on a single day [point prevalence] and over 12 months [period prevalence]; percentage of appropriate prescriptions). Meta-regression examined associations between antibiotic use, year of measurement and region. A total of 90 articles representing 78 studies from 39 countries with data between 1985–2019 were included. Pooled estimates of point prevalence and 12-month period prevalence were 5.2% (95% CI: 3.3–7.9; n = 523,171) and 62.0% (95% CI: 54.0–69.3; n = 946,127), respectively. Point prevalence varied significantly between regions (Q = 224.1, df = 7, p<0.001), and ranged from 2.4% (95% CI: 1.9–2.7) in Eastern Europe to 9.0% in the British Isles (95% CI: 7.6–10.5) and Northern Europe (95% CI: 7.7–10.5). Twelve-month period prevalence varied significantly between regions (Q = 15.1, df = 3, p = 0.002) and ranged from 53.9% (95% CI: 48.3–59.4) in the British Isles to 68.3% (95% CI: 63.6–72.7) in Australia. Meta-regression found no association between year of measurement and antibiotic use prevalence. The pooled estimate of the percentage of appropriate antibiotic prescriptions was 28.5% (95% CI: 10.3–58.0; n = 17,245) as assessed by the McGeer criteria. Year of measurement was associated with decreasing appropriateness of antibiotic use over time (OR:0.78, 95% CI: 0.67–0.91). The most frequently used antibiotic classes were penicillins (n = 44 studies), cephalosporins (n = 36), sulphonamides/trimethoprim (n = 31), and quinolones (n = 28).

available in the manuscript or in its Supplementary files.

**Funding:** MZR is supported by a National Health and Medical Research Council (https://www.nhmrc.gov.au/) Early Career Fellowship (APP1143941). JIW is supported by a National Health and Medical Research Elizabeth Blackburn Leadership Investigator Grant (APP1174021). The funders had no role in study design, data collection and analysis, decision to publish, or preparation of the manuscript.

**Competing interests:** The authors have declared that no competing interests exist.

## Conclusions

Coordinated efforts focusing on LTCFs are required to address antibiotic misuse in LTCFs. Our analysis provides overall baseline and regional estimates for future monitoring of antibiotic use in LTCFs.

## Introduction

Antibiotic use in long-term aged care facilities (LTCFs) continues to be a global concern, particularly amid rising antimicrobial resistance [1–3]. LTCF residents are at higher risk of infections due to age-related physiological changes, comorbidities, higher rates of hospitalisation, and close contact with other residents and facility staff [4]. Studies have highlighted that LTCF residents are often prescribed antibiotics inappropriately, more frequently and for longer than people in the community [4–10]. Overuse of antibiotics exposes LTCF residents to adverse events, and an increased risk for the development of antimicrobial resistance [11, 12].

The World Health Organization's (WHO) global strategy to curb antimicrobial resistance includes surveillance of antibiotic use [13]. The WHO's three tiered AWaRe classification groups antibiotics according to whether their use is preferred or should be restricted, providing a framework for stewardship programs. The classification's three tiers include: i) "Access" antibiotics of choice for each of the most common infections, ii) "Watch" antibiotics that are recommended only for specific, limited indications and are the "highest priority critically important" antibiotics; and iii) "Reserve" antibiotics that should only be used as a last resort.

Understanding patterns of antibiotic use is crucial for developing and planning policies to address inappropriate use. In response to concerns over high rates of inappropriate antibiotic use in LTCFs, survey programs have been initiated in the last decade to monitor antibiotic use in this setting. These include the Healthcare-Associated infections in Long-Term care (HALT) [14] and European Surveillance of Antimicrobial Consumption (ESAC) [15] surveys run by the European Centre for Disease Prevention and Control, and the Australian Aged Care National Antimicrobial Survey (acNAPS) [16]. Guidelines on monitoring of antibiotic use have been published [17–19], and there have been increasing efforts to develop and evaluate antimicrobial stewardship programs specifically for LTCFs [20–23]. However, a comprehensive summary of the available data on antibiotic use in LTCFs is not available. Two previous narrative international reviews on antibiotic use in LTCFs were published in 2012 [6] and 2019 [4], and reported data on antibiotic use up to 2011 and 2017, respectively. Thus, despite antibiotic surveillance being a core component of the WHO's global strategy to addressing antibiotic resistance, there is an absence of recent global estimates of antibiotic use in LTCFs. We aimed to address this gap. Our aims were to: i) undertake the first meta-analysis to quantitatively summarise key indicators of antibiotic use and appropriateness of use in LTCFs; ii) to explore temporal and regional trends and differences in antibiotic use using meta-regression; and iii) to summarise the most frequently used antibiotic classes overall and by geographic region.

## Materials and methods

The protocol for this systematic review and meta-analysis was registered with PROSPERO (ID: CRD42018107125l; S1 File) and reported according to the Preferred Reporting Items for Systematic Reviews and Meta-analyses (PRISMA; checklist in S2 File) guideline.

## Information sources and search strategy

We searched the bibliographic databases Medline (via OvidSP and PubMed) and EMBASE (via OvidSP) for original research articles reporting rates of antibiotic use in LTCFs published in English between 1990–2021. Studies could report data prior to 1990. The search strategy used a combination of keywords and subject headings on 'long-term care facilities' and 'antibiotics'. The full search strategy is provided in the S3 File. Reference lists of included articles were hand searched for further relevant citations. Authors of published papers were contacted for additional details when required.

## Eligibility criteria and study selection

Articles reporting rates of systemic antibiotic use in LTCFs for the aged were eligible, including intervention studies. Examples of antibiotic use rates are: the percentage of residents on an antibiotic (prevalence); number of antibiotic courses per 1000 resident days; days of therapy (DOT) per 1000 resident days; and the number of defined daily doses (DDD) per 1000 resident days. Studies not providing separate estimates for systemic antibiotic use were included if it was stated that topical use was a small proportion ($\leq$10%) of overall use. To ensure we were including facilities predominantly for older adults, we included studies in long-term aged care facilities, skilled nursing facilities, nursing homes, residential aged care facilities and assisted living facilities; but excluded studies in specialist LTCFs described as psychiatric, palliative, rehabilitation, for physically disabled people or those in hospitals. Studies only reporting rates of antibiotic use for specific infections were excluded.

Two reviewers from a panel of three (MR, CG, PG) independently screened the title and abstracts of papers generated by the literature search, after exclusion of duplicates, and then assessed potential full-text articles to determine eligibility. Disagreements in screening were resolved through discussion between reviewers until consensus was achieved.

## Data collection process and data items

Data were extracted independently by three authors (MR, PG, CG) and discrepancies were resolved through discussion. Multiple articles from the same study were grouped and data extracted for the study, rather than each article. For each study data on: the study country, sample size (number of LTCFs and residents), study design (point prevalence survey, longitudinal cohort study, intervention study), period of data collection (which could precede 1990), and the method used to measure antibiotic use (e.g. point prevalence survey, pharmacy supply databases, chart review) were extracted. Characteristics of LTCF residents recorded included their demographics (mean or median age, proportion of women) and prevalence of key conditions that may affect antibiotic use (dementia and catheter use). Data on the antibiotic use rate, appropriateness of use and most frequently used antibiotics were extracted. For studies with an intervention, the baseline antibiotic use rates were extracted. For appropriateness of antibiotic therapy, the assessment criteria used (e.g. McGeer criteria) were recorded as methods for assessing appropriateness vary, and can assess whether an antibiotic was required or whether the chosen antibiotic follows treatment guidelines. When available, the three most frequently used antibiotics and/or antibiotic classes were also recorded.

Countries were grouped into regions for analysis. Regions consisted of continents, with Europe divided further into five sub-regions broadly following European Union terminology: British Isles, Western Europe, Southern Europe, Northern Europe, Eastern Europe [24].

We grouped antibiotics into antibiotic classes using the World Health Organisation's Anatomical Therapeutic Chemical Classification System (ATC) level 3 codes [25]. These included: tetracyclines (ATC: J01A), pencillins (J0C), other beta-lactams (including cephalosporins;

J01D), sulfonamides and trimethoprim (J01E), macrolides (J01F), quinolones (J01M), and other classes (includes only methenamine and nitrofurantoin; J01X).

### Assessment of study quality

Two reviewers (MR, PG) independently assessed the quality of each study using the Joanna Briggs Institute Critical Appraisal Tool for Prevalence Studies [26]. The tool includes nine assessment criteria regarding the sampling strategy, sample description, methods of measurement, and statistical analysis. An overall rating of quality was then assigned to each study based on the number of criteria with a 'yes' score. Studies were rated as *good quality* when they met 7–9 of the criteria; *fair quality* for 5–6 of the criteria, and *poor quality* when they met <5 criteria.

### Synthesis of results

The available data allowed us to conduct meta-analyses for three antibiotic use measures: point prevalence, 12-month period prevalence, and the proportion of appropriate prescriptions as assessed against the McGeer criteria. There was a lack of reporting of measures of dispersion (e.g. standard deviation) and denominator sample sizes for other outcomes (DDD/1000 days, DOT/1000 days, number of courses/1000 days, other appropriateness criteria) to allow meta-analysis. Thus, a narrative synthesis was used for these outcomes. Point prevalence was defined as the percentage of residents on an antibiotic on one day. Twelve-month period prevalence was defined as the percentage of residents who used an antibiotic over 12 months and was extracted from studies measuring use over 12-months (i.e. 12-month period prevalence was not extrapolated from studies measuring prevalence over shorter or longer periods).

We conducted sub-group meta-analyses by regions to generate pooled estimates of antibiotic use by region and overall. We used random effects models (i.e. random effects models for each region and to generate an overall estimate) with a pooled estimate of $Tau^2$ ($T^2$). Pooling $Tau^2$ overcomes the limitation of underestimating the variance in subgroups with fewer studies. Heterogeneity among study estimates within regions and between regions were examined by visually inspecting forest plots, the Q-statistic, Higgins' $I^2$ and $T^2$ [27]. Publication bias was assessed using funnel plots and the Egger's regression test, which is a measure of funnel plot asymmetry [28]. Duval and Tweedie's Trim and Fill method [29] was used to correct for publication bias in the meta-analyses when it was present.

We further explored the heterogeneity in estimates of antibiotic use and appropriateness using meta-regression. Meta-regression was fitted using restricted maximum likelihood estimation with a Knapp-Hartung adjustment. We examined the association between the three outcomes (point prevalence, 12-month period prevalence estimates, appropriateness), and the year of data collection and region. Since antibiotic use is higher during influenza season, we adjusted the point prevalence model for whether data collection took place during the influenza season [30]. When estimates were available for only one country in a region, we used country names. We were not able to include independent variables for resident demographics or health conditions (dementia and catheter use prevalence) in the models as they were not reported with enough consistency in the studies. However, the demographic details reported in the studies are provided in Supplementary files. An estimate of $R^2$ was used to estimate the proportion of between study variance explained by the model. Comprehensive Meta-Analysis 3.0 was used to perform all analyses [31] and R Studio was used to generate forest plots [32].

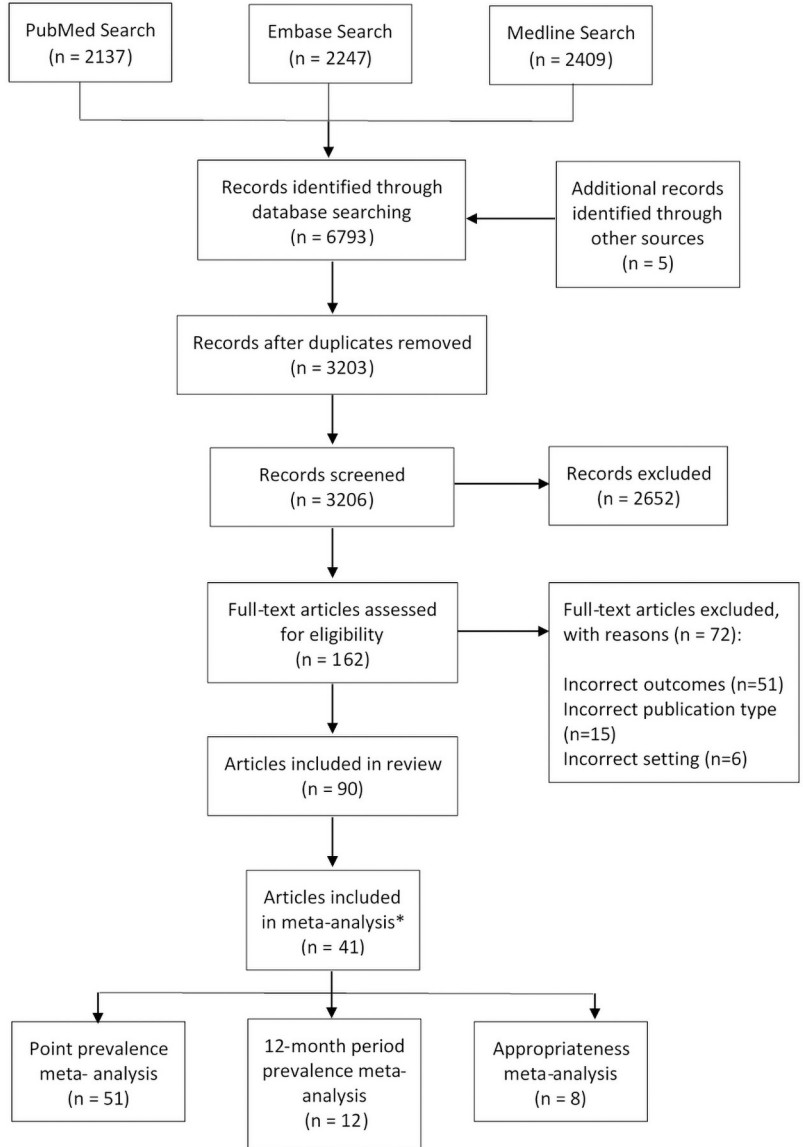

**Fig 1. Flowchart of search strategy and screening results.** *Some studies contributed more than one estimate to the meta-analyses.

## Results

### Study characteristics

A total of 90 articles and reports were included [7–10, 15, 18, 30, 33–109], representing 78 individual studies (Fig 1). Study characteristics are presented in Table 1. There were 29 studies from Europe and the British Isles, 32 from North America (24 from the US, 9 from Canada, and one from both US and Canada), 12 from Australia, and four from Asia (two from Singapore and two from Japan); plus one study reporting data from Australia and the Netherlands. Study sample sizes had a median of 39 LTCFs (interquartile range: 9–174). The majority of studies reported the outcome percent of residents using an antibiotic (period and/or point prevalence; n = 49), followed by percentage of appropriate prescriptions (n = 25), number of

**Table 1. Characteristics of included studies measuring the antibiotic use rates in long-term care facilities.**

| Author, year | Country | Number of facilities | Data collection year/s | Study design | Antibiotic use data source | Outcomes[a] | Overall quality rating[b] |
|---|---|---|---|---|---|---|---|
| Alberg, 2017 [33] | Norway | 540 | 2016 | Cross-sectional study | Point prevalence survey | 1, 5 | Poor (4/9) |
| acNAPS, 2016 [35] | Australia | 186 | 2015 | Cross-sectional study | Point prevalence survey | 1, 5 | Good (8/9) |
| acNAPS, 2017 [36, 121] | Australia | 287 | 2016 | Cross-sectional study | Point prevalence survey | 1, 5 | Good (8/9) |
| acNAPS, 2018 [37] | Australia | 292 | 2017 | Cross-sectional study | Point prevalence survey | 1, 5 | Good (8/9) |
| acNAPS, 2019 [38] | Australia | 407 | 2018 | Cross-sectional study | Point prevalence survey | 1, 5 | Good (8/9) |
| acNAPS, 2020 [39] | Australia | 568 | 2019 | Cross-sectional study | Point prevalence survey | 1 | Good (7/9) |
| Barney, 2019 [34] | United States | 4 | 2016–2017 | Retrospective cohort | Pharmacy database | 1, 2 | Good (7/9) |
| Benoit, 2008 [40] | United States | 73 | 2001–2002 | Retrospective cohort | Chart review | 1, 3 | Good (7/9) |
| Blix, 2007 [41] | Norway | 133 | 2003 | Retrospective cohort | Antibacterial sales database | 4 | Good (8/9) |
| Blix, 2010 [125] | Norway | 44 | 2006 | Cross-sectional study | Point prevalence survey | 1 | Good (7/9) |
| Boivin, 2013 [42] | France | 18 | 2012 | Retrospective cohort | Chart review; point prevalence survey | 1, 5 | Good (8/9) |
| Cowan, 2016 [43] | Australia | 2 | 2014 | Retrospective cohort | Chart review | 1, 5 | Poor (1/9) |
| Daneman, 2011 [116] | Canada | 363 | 2009 | Retrospective cohort | Pharmacy database | 1 | Good (7/9) |
| Daneman, 2013 [7] | Canada | 630 | 2010 | Retrospective cohort | Pharmacy database | 1 | Good (7/9) |
| Daneman, 2015 [44] | Canada | 607 | 2010–2011 | Retrospective cohort | Pharmacy database | 2 | Good (8/9) |
| Daneman, 2017 [45] | Canada | 628 | 2014 | Retrospective cohort | Pharmacy database | 1 | Good (7/9) |
| Drinka, 2004 [46] | United States | 1 | 1996–2002 | Retrospective cohort | Pharmacy database | 1 | Fair (6/9) |
| Eikelenboom-Boskamp, 2019 [47] | Netherlands | 25, 44 | 2010–2017 | Cross-sectional study | Point prevalence survey | 1 | Good (8/9) |
| ESAC-1 [15, 48–51] | 21 European countries | 323 | 2009 | Cross-sectional study | Point prevalence survey | 1, 4, 5[c] | Good (8/9) |
| ESAC-2 [51, 52] | Northern Ireland, Finland | 30 & 9 | 2010 | Cross-sectional study | Point prevalence survey | 1, 5 | Good (8/9) |
| ESAC-3 [52] | Northern Ireland | 30 | 2011 | Cross-sectional study | Point prevalence survey | 1, 5 | Good (8/9) |
| Fagan, 2012 [53] | Norway | 10 | 2007–2008 | Retrospective Cohort | Health record data | 1, 4, 5 | Good (7/9) |
| Felsen, 2020 [54] | United States | 6 | 2014–2018 | Intervention study | Pharmacy database | 2 | Poor (4/9) |
| Fleet, 2014 [55] | England | 30 | 2010–2011 | Intervention study | Chart review | 1, 4, 5 | Fair (6/9) |
| Gillespie, 2015 [56] | Wales | 10 | 2010–2012 | Retrospective Cohort | Chart review | 1, 3 | Good (7/9) |
| HALT-1 [57–60] | 28 European countries | 676[d] | 2010 | Cross-sectional study | Point prevalence survey | 1, 5[e] | Good (8/9) |
| HALT-2 [18, 63, 64] | 19 European countries | 1051[d] | 2013 | Cross-sectional study | Point prevalence survey | 1 | Good (8/9) |
| HALT-3 [61, 62, 65] | 24 European countries | 1,788[d] | 2016–2017 | Cross-sectional study | Point prevalence survey | 1, 5 | Good (8/9) |
| Heudorf, 2012 [66] | Germany | 40 | 2011 | Cross-sectional study | Point prevalence survey | 1 | Good (7/9) |
| Ishikane, 2020 [67] | Japan | 6 | 2016 | Retrospective cohort | Pharmacy database | 4 | Fair (5/9) |
| Jump, 2012 [68] | United States | 1 | 2006–2010 | Retrospective Cohort | Pharmacy database | 2 | Fair (6/9) |
| Kabbani, 2019 [69] | United States | 12 | 2016 | Retrospective Cohort | Pharmacy database | 2 | Good (8/9) |
| Katz, 1990 [70] | United States | 2 | 1985 | Prospective cohort | Chart review; observation | 3, 5 | Fair (6/9) |
| Lee, 1992 [71] | United States | 7 | 1989 | Prospective cohort | Chart review; point prevalence survey | 1, 5 | Fair (6/9) |
| Lee, 1996 [72] | United States | 1 | Not reported | Prospective cohort | Chart review; point prevalence survey | 1, 5 | Fair (5/9) |
| Loeb, 2005 [73] | United States, Canada | 24 | 2001–2003 | Intervention study | Chart review | 3 | Fair (6/9) |
| Marquet, 2015 [74] | France | 52, 74 | 2011–2013 | Retrospective Cohort | Pharmacy database | 4 | Fair (5/9) |

(*Continued*)

**Table 1.** (Continued)

| Author, year | Country | Number of facilities | Data collection year/s | Study design | Antibiotic use data source | Outcomes[a] | Overall quality rating[b] |
|---|---|---|---|---|---|---|---|
| Marra, 2017 [8] | Canada | 381 | 2007–2014 | Retrospective Cohort | Pharmacy database | 3, 4 | Good (8/9) |
| Mayne, 2018 [9] | Australia | 5 | 2015–2016 | Prospective cohort | Chart review | 1 | Fair (6/9) |
| Monette, 2007 [75] | Canada | 8 | 2001–2003 | Intervention study | Pharmacy database; chart review | 1, 5 | Poor (4/9) |
| Montgomery, 1995 [76] | Canada | 100 | 1986 | Retrospective cohort | Chart review | 1, 5 | Good (7/9) |
| Moro, 2007 [77] | Italy | 49 | 2001 | Cross-sectional study | Point prevalence survey | 1 | Good (7/9) |
| Mylotte, 1999 [78] | United States | 4 | 1996–1998 | Prospective cohort | Survey | 3 | Fair (5/9) |
| Mylotte, 2005 [111] | United States | 11 | 2003 | Retrospective cohort | Survey | 2 | Fair (5/9) |
| Natsch, 1998 [79] | Netherlands | 6 | 1995 | Retrospective Cohort | Pharmacy database | 4 | Fair (4/9) |
| Olsho, 2013 [80] | United States | 12 | 2011 | Prospective cohort | Chart review | 3, 5 | Good (7/9) |
| Pakyz, 2010 [81][f] | United States | 1174 | 2004 | Retrospective cohort | Chart review | 1 | Good (8/9) |
| Pluss-Suard, 2020 [82] | Switzerland | 23 | 2011–2016 | Prospective cohort | Pharmacy database | 4 | Fair (5/9) |
| Raban, 2020 [30] | Australia | 68 | 2014–2017 | Retrospective cohort | Health record data | 1, 2, 3 | Good (9/9) |
| Rahme, 2016 [83] | United States | 1 | 2012–2013 | Prospective cohort | Pharmacy database | 4 | Fair (6/9) |
| Roukens, 2017 [84] | Netherlands | 31 | 2012–2014 | Point prevalence; Retrospective cohort | Survey; chart review | 4 | Fair (5/9) |
| Rummukainen, 2009 [85] | Finland | 29 | 2004–2006 | Cross-sectional study | Point prevalence survey | 1 | Fair (6/9) |
| Rummukainen, 2013 [86] | Finland | 263 | 2011 | Cross-sectional study | Point prevalence survey | 1 | Good (8/9) |
| Saxena, 2019 [87] | Canada | 87,947 (residents) | 2016 | Retrospective cohort | Pharmacy database | 1 | Good (7/9) |
| Selcuk, 2018 [88] | Singapore | 4 | 2008 | Retrospective cohort | Chart review | 1, 2, 4 | Fair (5/9) |
| Selcuk, 2019 [89] | Singapore | 9 | 2017 | Cross-sectional study | Point prevalence survey | 1, 4 | Fair (5/9) |
| Sloane, 2014 [90] | United States | 4 | 2010–2012 | Prospective cohort | Chart review | 3 | Fair (5/9) |
| Sloane, 2019 [91] | United States | 14 | 2015–2017 | Prospective cohort | Chart review | 3 | Fair (6/9) |
| Sloane, 2020 [91] | United States | 27 | 2015–17 | Intervention study | Chart review by nursing home staff | 3 | Fair (5/9) |
| Sluggett, 2020 [92] | Australia | 3218 | 2005–2006; 2010–2011; 2015–2016 | Repeated cross-sectional study | National pharmaceutical claims data | 1, 4 | Good (9/9) |
| Smith, 2013 [94] | Australia | 29 | 2011 | Cross-sectional study | Point prevalence survey | 1, 5 | Good (8/9) |
| Smith, 2020 [93] | United Kingdom | 135 | 2016–2017 | Retrospective cohort | Pharmacy database; electronic records | 3[g] | Good (8/9) |
| Song, 2021 [95] | United States | 29 | 2016 | Retrospective cohort | Invoice data | 2, 3 | Good (9/9) |
| Stepan, 2018 [96] | Slovenia | 80 | 2016 | Cross-sectional study | Point prevalence survey | 1 | Good (7/9) |
| Stuart, 2012 [98] | Australia | 5 | 2011 | Cross-sectional study | Point prevalence survey | 1, 5 | Fair (5/9) |
| Stuart, 2015 [97] | Australia | 2 | 2012 | Prospective cohort | Chart review | 2, 5 | Poor (4/9) |
| Sundvall, 2015 [99] | UK | 7481 (residents) | 2011 | Retrospective cohort | Health record data | 1 | Good (7/9) |
| Takito, 2020 [100] | Japan | 1 | 2013–2017 | Intervention study | Chart review | 3[g] | Poor (1/9) |
| Taxis, 2017 [101] | Australia, Netherlands | 26 & 6 | 2009 | Retrospective cohort | Pharmacy database | 1 | Good (7/9) |
| Temime, 2018 [102] | France | 13 | 2014–2015 | Intervention study | LTCF database | 4 | Poor (4/9) |
| Thompson, 2016 [103, 113] | United States | 9 | 2013–2014 | Point prevalence | Survey | 1, 5 | Poor (4/9) |
| Thompson, 2021 [104] | United States | 161 | 2017 | Point prevalence | Survey | 1 | Good (8/9) |
| Thornley, 2019a [10] | United Kingdom | 341,536 (residents) | 2016–2017 | Retrospective cohort | Pharmacy database | 1 | Good (7/9) |

(*Continued*)

**Table 1.** (Continued)

| Author, year | Country | Number of facilities | Data collection year/s | Study design | Antibiotic use data source | Outcomes[a] | Overall quality rating[b] |
|---|---|---|---|---|---|---|---|
| Thornley, 2019b [105] | United Kingdom | 644 | 2017 | Cross-sectional study | Point prevalence survey | 1 | Good (7/9) |
| van Buul, 2015 [106] | Netherlands | 10 | 2012–2013 | Intervention study | Pharmacy database | 3 | Good (7/9) |
| Warren, 1991 [107] | United States | 52 | 1985–1986 | Prospective cohort | Chart review | 1, 3, 5 | Fair (5/9) |
| Wu, 2015 [108] | Canada | 17 | 2011–2012 | Retrospective cohort | Chart review | 2 | Good (7/9) |
| Zimmerman, 2014 [109] | United States | 12 | 2011 | Intervention study | Chart review | 3 | Fair (6/9) |

acNAPS: Aged Care National Antimicrobial Prescribing Survey, Australia; ESAC: European Surveillance of Antimicrobial Consumption; HALT: Healthcare-Associated Infections in Long-Term Care Facilities Project, Europe.

[a]Outcome 1 is percentage of residents on an antibiotic; 2 is days of therapy per 1000 residents; 3 is courses per 1000 resident days; 4 is defined daily doses per 1000 resident days; 5 is percentage of appropriate antibiotic prescriptions.

[b]Study quality assessed based on the Joanna Briggs Institute Critical Appraisal Tool for Prevalence Studies. The score in the brackets is the total number of criteria met.

[c]Subset of facilities from Northern Ireland provided the percentage of appropriate antibiotic prescriptions.

[d]Number of participating general nursing homes, residential homes, and mixed long-term care facilities. Other facility types reported in the HALT surveys excluded here are psychiatric long-term care facilities, long-term care facilities for the mentally disabled, long-term care facilities for the physically disabled, rehabilitation centres, palliative care centres, and 'other' long-term care facilities.

[e]Subset of facilities from Italy.

[f]Pakyz et al. report on results of the National Nursing Home Survey, conducted by the Centers for Disease Control and Prevention's National Center for Health Statistics.

[g]Smith reported as number of prescriptions per resident year. Takito reported as number of prescription per 100 residents per month.

courses per 1000 resident days (n = 17), DDD/1000 resident days (n = 15), and DOT/1000 days (n = 12). Thirty-three studies reported multiple outcomes. Most studies (n = 51) used surveys or chart review to measure antibiotic use, and 27 studies used electronic records or databases of medication supply.

## Assessment of study quality

We assessed 45 studies to be of good quality, 25 as fair, and only eight studies to be of poor quality (Table 1, and detailed assessment in S4 File). Criteria 4 and 8 were most frequently assessed as not being met (both scored as 'no' in 46 studies) across the studies. Criterion 4 requires descriptions of the study setting and subjects; and criterion 8 requires appropriate statistical analysis, including reporting of denominators, confidence intervals or standard errors.

## Point prevalence estimates of antibiotic use in LTCFs

Meta-analysis was conducted on a total of 123 point-prevalence estimates from 37 countries between 1985 and 2019. Resident characteristics were available for 71 of the estimates, and eligibility criteria for residents were similar across these studies, primarily following those used in the HALT and ESAC programs (S5 File). The percentage of residents receiving an antibiotic on a single day ranged from 0.7% to 17.3%. The pooled estimate, taking into account subgroups, was 5.2% (95% CI: 3.3–7.9%; n = 523,171; $I^2$ = 98.6; $T^2$ = 0.162) with a significant test for differences between regions (Q = 224.1, df = 7, p<0.001). Table 2 shows the pooled estimates for regions and detailed heterogeneity statistics are shown in S6 File. The Q-statistics for within region heterogeneity indicated significant variation between studies within regions (S6 File), for all regions except Singapore (which had only two studies). The $I^2$ values for each region indicated that the majority of the heterogeneity was likely due to real variation in estimates.

**Table 2. Pooled estimates from meta-analysis of point prevalence of antibiotic use by region.**

| Region | Number of estimates | Range of point prevalence estimates | Pooled point prevalence (95% CI) | $I^2$ | $T^2$ |
|---|---|---|---|---|---|
| Singapore | 2 | 2.33, 2.97 | 2.6 (1.4, 4.7) | 0.0 | 0.000 |
| Australia | 7 | 5.47, 8.95 | 7.2 (5.4, 9.5) | 94.6 | 0.026 |
| British Isles | 22 | 5.53, 12.7 | 9.0 (7.7, 10.4) | 92.8 | 0.063 |
| Eastern Europe | 27 | 0.73, 11.3 | 2.3 (1.9, 2.7) | 95.6 | 0.493 |
| Northern Europe | 24 | 2.72, 17.3 | 9.1 (7.8, 10.6) | 97.4 | 0.224 |
| Southern Europe | 16 | 0.79, 12.2 | 4.9 (4.0, 6.1) | 97.8 | 0.249 |
| Western Europe | 21 | 1.15, 6.10 | 3.2 (2.7, 3.8) | 97.3 | 0.224 |
| North America | 4 | 5.86, 11.1 | 7.2 (4.7, 10.9) | 98.1 | 0.055 |
| **OVERALL** | **123** | **0.73, 17.3** | **5.2 (3.3, 7.9)** | **98.6** | **0.077**[a] |

[a]Within region Tau$^2$ pooled across regions.

Sensitivity analysis removing one [103] study of poor quality gave a pooled estimate of 5.1% (95% CI: 3.3–7.8%; $I^2$: 98.6; $T^2$: 0.156). The Egger's test showed potential publication bias (p = 0.003) so we employed the Duval and Tweedie's trim and fill method. A total of 18 studies were trimmed, resulting in an adjusted point prevalence estimate of 6.2% (95% CI: 5.6–6.8%).

Meta-regression examined the association between point prevalence estimates of the percentage of residents on an antibiotic, and year of measurement and geographic region, while adjusting for whether measurement took place during influenza season (Table 3). There was no significant association with the year of measurement. All regions, were more likely to have LTCF residents on an antibiotic on the day of survey compared to Eastern Europe, with the exception of Singapore (Table 3). The model explained 56% of between study variance.

## Period prevalence estimates of antibiotic use

There was a total of 30 period prevalence estimates from 11 countries between 1985 and 2017. Of these, 19 were estimates of the percentage of residents who used an antibiotic over 12-months (12-month period prevalence) from nine countries between the years 1985–2017.

**Table 3. Meta-regression of point prevalence estimates of antibiotic use in long-term care facilities (N = 123).**

| Independent variable | | No. of estimates | Odds ratio (95% CI) | p-value |
|---|---|---|---|---|
| Year | | 123 | 0.98 (0.96, 1.01) | 0.284 |
| Measured during flu season | | | | |
| | No | 105 | 1.00 | |
| | Yes | 18 | 1.01 (0.67, 1.53) | 0.968 |
| Region | | | | |
| | Eastern Europe | 27 | 1.00 | |
| | Northern Europe | 24 | 4.16 (3.04, 5.69) | <0.001 |
| | Western Europe | 21 | 1.43 (1.03, 1.97) | 0.032 |
| | Southern Europe | 16 | 2.18 (1.52, 3.14) | <0.001 |
| | British Isles | 22 | 4.24 (3.03, 5.94) | <0.001 |
| | Australia | 7 | 3.53 (1.90, 6.55) | 0.0001 |
| | Singapore | 2 | 1.17 (0.50, 2.71) | 0.716 |
| | North America | 4 | 3.43 (1.76, 6.70) | 0.0004 |

$R^2$: 0.56 (estimate of proportion of between-study variance explained by model).

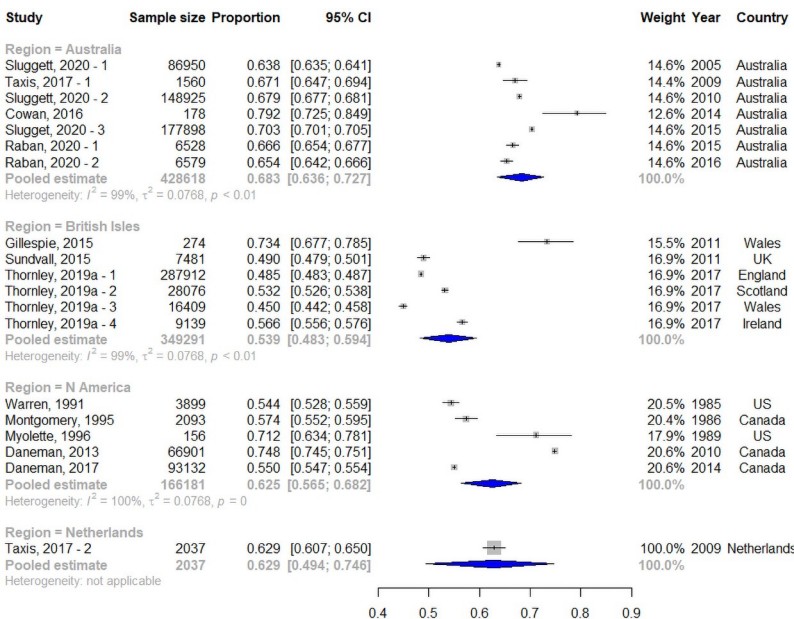

**Fig 2. Meta-analysis by region of twelve-month period prevalence of antibiotic use.** CI is confidence interval. Markers for individual studies are proportional to the studies' weight in generating the region estimate.

Resident characteristics were available for 11 of these studies (S7 File). The 12-month period prevalence ranged from 45.0% to 79.2% of residents. Meta-analysis, taking into account subgroups, gave a pooled estimate of 62.0% (95% CI: 54.0–69.3%; n = 946,127; $I^2$: 100%; $T^2$: 0.077) of residents using an antibiotic annually, with statistically significant differences between regions (Q = 15.1, df = 3, p = 0.002). Fig 2 shows the pooled estimates for each region. $I^2$ values indicated that almost all variation was likely due to real differences in estimates (S6 File).

Sensitivity analysis, removing two studies [43, 110] of poor quality, gave a pooled estimate of 61.0% (95% CI: 53.767.9%; $I^2$: 100%; $T^2$: 0.077). The Egger's test showed no significant publication bias (p = 0.714).

Meta-regression showed year of measurement was not significantly associated with 12-month period prevalence (Table 4). Of the regions, Australia had higher odds of residents being on an antibiotic over a 12-month period compared to the British Isles (Table 4). The model explained 24% of the total between study variance.

Nine other studies estimated the percentage of residents on an antibiotic over varying time periods of six months or less, and all were of fair or good quality (S8 File) [9, 34, 40, 47, 48, 71, 75, 81, 88].

**Table 4. Meta-regression of 12-month prevalence estimates of antibiotic use in long-term care facilities (N = 19).**

| Independent variable | No. of estimates | Odds ratio (95% CI) | p-value |
|---|---|---|---|
| Year | 19 | 1.01 (0.98, 1.03) | 0.501 |
| Region | | | |
| British Isles | 6 | 1.00 | |
| Netherland | 1 | 1.51 (0.66, 3.47) | 0.300 |
| Australia | 7 | 1.76 (1.13, 2.75) | 0.017 |
| North America | 5 | 1.51 (0.79, 2.86) | 0.188 |

$R^2$: 0.24 (estimate of proportion of between-study variance explained by model).

## Other estimates of antibiotic use

A total of 14 studies reported a range of mean antibiotic use from 2.7 to 237 DOT/1000 bed days [30, 34, 44, 54, 68, 69, 78, 88, 91, 95, 97, 106, 108, 111]. Fourteen studies reported a range from 2.1 to 13.0 antibiotic courses per 1000 resident days [8, 30, 40, 56, 70, 73, 80, 90, 93, 95, 107, 109, 111, 112]. Finally, 15 studies reported a range of 38.2 to 148 DDD/1000 bed days [8, 41, 53, 55, 67, 74, 79, 82–84, 88, 89, 92, 102, 106]. Details are provided in S8 File.

## Appropriateness of antibiotic therapy

Nine studies, conducted between 2010 and 2018, assessed the appropriateness of antibiotic treatment against the McGeer criteria [35–38, 55, 60, 97, 98, 113], which assess whether antibiotic treatment was warranted based on the presence of infection symptoms. Eight studies included courses for all infections [35–38, 55, 60, 97, 98], and one study assessed appropriateness for UTIs [113], which was excluded from the meta-analysis. The percentage of appropriate courses ranged between 9.5% to 60.3% in individual studies, conducted between 2010 and 2018 in three countries (S9 File shows the resident characteristics reported it seven of the studies). Meta-analysis, taking into account sub-groups by country, gave a pooled estimate of appropriateness of 28.5% (95% CI: 10.3–58.0; n = 17,245; $I^2$: 99.6%; $Tau^2$: 0.858) with differences between regions not significant as assessed by the Q-statistic (Q = 3.57, df = 2, p = 0.167). S6 File shows the detailed heterogeneity statistics. Fig 3 shows pooled estimates for each region. Sensitivity analysis removing two studies of poor quality [97, 113] yielded a pooled estimate of 28.0% (95% CI: 9.9–57.9%; $I^2$: 99.7%; $T^2$: 0.869). The Egger's test showed no significant publication bias (p = 0.352).

Meta-regression showed the percentage of appropriate courses decreased every year and Australia and Italy had higher rates of appropriate courses compared to England (Table 5). The model explained 85% of the between study variance.

Studies also applied other methods to assess appropriateness (n = 16), including the Loeb Minimum Criteria; local and national algorithms and guidelines; microbiology laboratory results; urinalysis; and expert consensus. Estimates of the percentage of appropriate prescriptions ranged from 5.6% to 88.5% when assessing prescribing for all infections; 4% to 81.3% respiratory tract infections; 13% to 59.3% for skin and soft tissue infections; and 18% to 97.3% for urinary tract infections. Details are provided in S8 File.

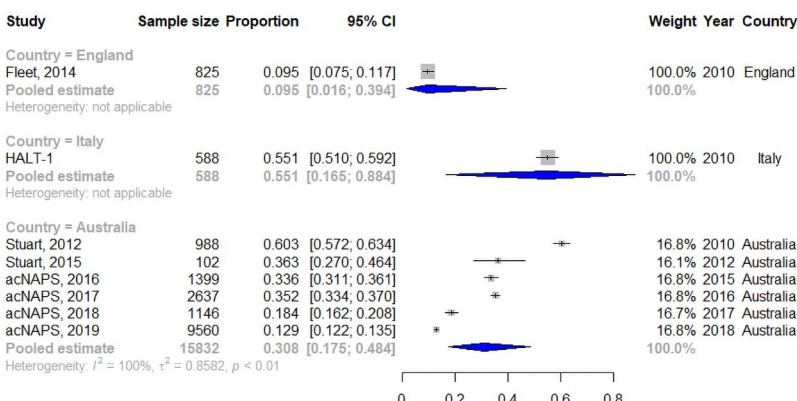

**Fig 3. Meta-analysis by region of the percent of appropriate prescriptions according to McGeer criteria.** CI is confidence interval. Markers for individual studies are proportional to the studies' weight in generating the region estimate.

**Table 5. Meta-regression of the percentage of appropriate [a] antibiotic courses in long-term care facilities (N = 8).**

| Independent variable | No. of estimates | Odds ratio (95% CI) | p-value |
|---|---|---|---|
| Year | 8 | 0.78 (0.67, 0.91) | 0.0112 |
| Country | | | |
| England | 1 | 1.00 | |
| Australia | 6 | 13.64 (3.47, 53.52) | 0.0061 |
| Italy | 1 | 11.75 (2.63, 52.58) | 0.0103 |

$R^2$: 0.85 (estimate of proportion of between-study variance explained by model).

[a]Appropriateness was assessed using the McGeer criteria.

## Most frequently used antibiotics

There were 59 studies reporting on the most frequently used antibiotics or antibiotic classes. Fig 4 shows the three most frequently used classes of antibiotics reported in each study, grouped by region. Overall, the three most frequently reported antibiotic classes used were penicillins (n = 44 studies), cephalosporins (n = 36), and sulphonamides/trimethoprim (n = 31). Quinolones were among the three most common classes used in Western Europe (n = 3 studies), Southern Europe (n = 2), Eastern Europe (n = 2), North America (n = 14), and Asia (n = 4). Macrolides were among the three most common antibiotic classes in only four studies. There were nine studies that included data after 2017, the year when the WHO AWaRe classification was introduced. Quinolones, on the WHO's Watch List, were still among the top three antibiotics used in Asia, Southern Europe in North America from 2017 onwards.

## Discussion

In this systematic review and meta-analysis, we compiled the international literature on rates of systemic antibiotic use, appropriateness of use, and types of antibiotics used in LTCFs. Our review spans 34 countries across eight geographic regions between 1985 and 2019. We found significant variation between geographic regions in the point prevalence (average pooled estimate: 5.2% (95% CI: 3.3–7.9%)) and 12-month period prevalence (pooled estimate: 62.0% (95% CI: 54.0%-69.3%)) estimates of residents on an antibiotic. There was no significant change in the prevalence of use across all regions over time. The percentage of appropriate prescriptions (pooled average estimated: 28.5% (95% CI: 10.3–58.0)), as assessed by the McGeer criteria, also differed between regions, and decreased over time though this result is largely based on data from Australia. There were also regional differences in the most frequently used antibiotic classes, with quinolones being a common class in five of seven regions.

Our analysis of the point prevalence estimates of antibiotic use in LTCFs indicated that use was higher in the British Isles, followed by Northern Europe, and Australia compared to other regions. Twelve-month period prevalence estimates were available for only four regions, and antibiotic use was highest in Australia. The regions we identified as having the highest antibiotic use in LTCFs are broadly consistent with the regions reported to have high national levels of antibiotic consumption. An analysis of global antibiotic consumption, based on national pharmaceutical sales data between 2000 and 2010, reported national antibiotic consumption was highest in Australia, Ireland, and the United Kingdom compared to other European countries [2]. In addition, data from the European Surveillance of Antibiotic Consumption from 2017 also show that national antibiotic consumption in Australia and the British Isles is above the Organisation for Economic Cooperation and Development (OECD) country average

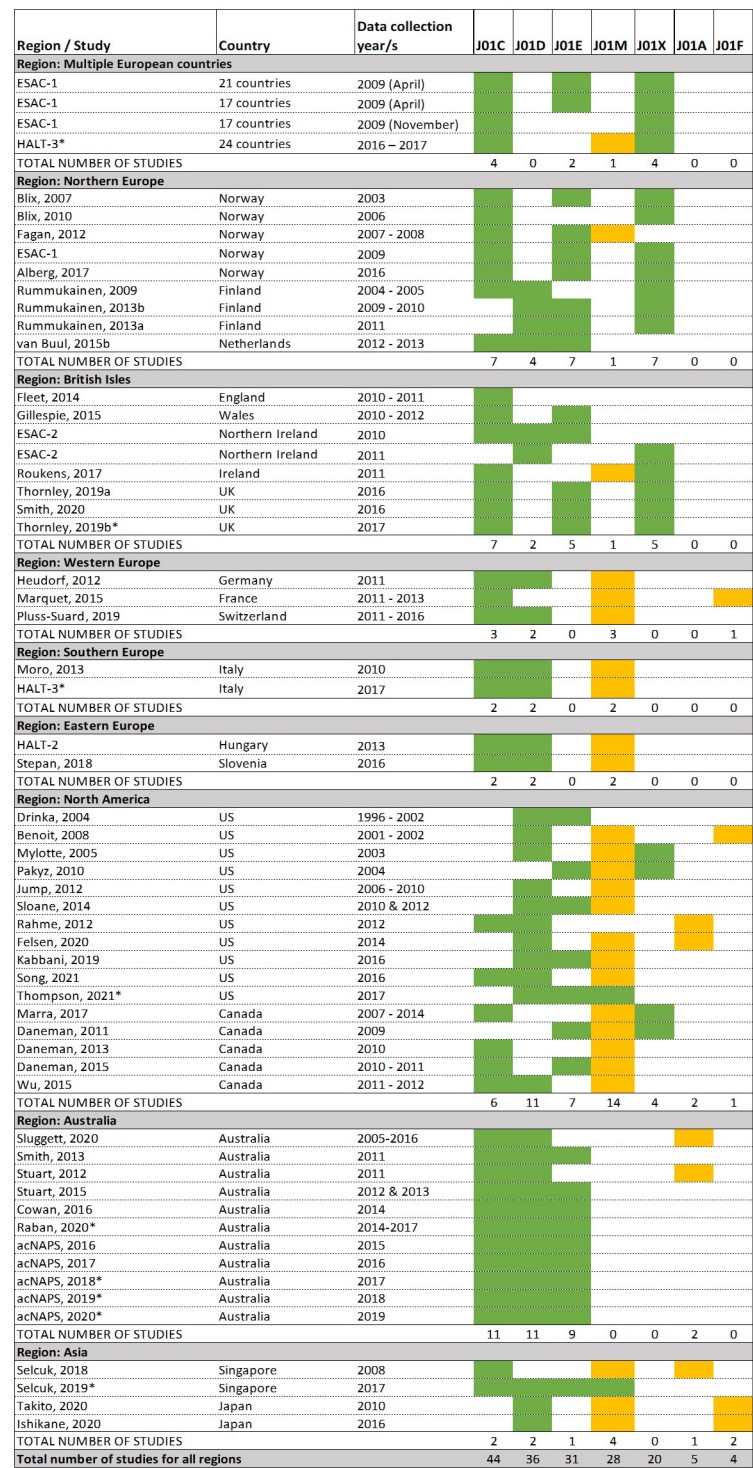

**Fig 4. Classes of three most frequently used antibiotics reported by studies.** J01C is penicillins; J01D is cephalosporins; J01M is quinolones; J01X is other classes (includes only methenamine and nitrofurantoin); J01A is tetracyclines; J01F is macrolides. Classes J01M, J01A and J01F are on the World Health Organization's (WHO) AWaRe 'Watch List' and should be targeted for reduced use due to high resistance potential. Studies including data from 2017 or later are indicated with an asterisk. The WHO AWaRe list was first published in 2017.

[114]. In contrast, national antibiotic consumption in countries of Northern Europe is well below the average consumption of OECD countries [2, 114], however our analysis suggests rates of antibiotic use in Northern European LTCFs are high compared to other regions. The higher rates of antibiotic use in LTCFs, despite low national antibiotic consumption, demonstrate the importance of examining antibiotic use by health care sector to identify where targeted interventions are required.

Variation in antibiotic use between regions could be attributable to a number of factors. Firstly, LTCF resident characteristics can impact antibiotic use and may vary between countries and regions. We extracted the available data on resident characteristics reported in studies (S5, S7, and S9 Files). An examination of the available (n = 71) resident profiles for studies included in our point prevalence meta-analysis reveals these were generally consistent (S5 File). For example, the age of residents in studies included in our meta-analyses of point prevalence showed that between 40–60% of residents were aged >85 years. A few studies had younger populations, particularly in Eastern Europe; however these represented the minority of all studies (3 of 12 in Eastern Europe with available data). Secondly, differences in health systems and models of care could also account for between region variation in antibiotic use. By using regions in our subgroup analysis, some of these differences can be accounted for, albeit in an imperfect way. A more detailed analysis of models of care or health system factors that lead to lower antibiotic use in LTCFs was beyond the scope of this analysis; however recent systematic reviews and analyses of facility level variation in antibiotic use elucidate some of these factors and strategies that may be effective in reducing use [20, 30, 45, 115]. Finally, differences in prescribing practices and LTCF policies in infection treatment and control have an impact on antibiotic use. This has been highlighted in research examining antibiotic use differences between LTCFs within countries and states, as well as those within single aged care provider networks [7, 10, 30, 115].

There were also regional differences in the types of antibiotics used. Quinolones were one of the three most frequently prescribed classes of antibiotics in five of the eight regions, particularly in studies from North America (14 of 16 studies in that region [7, 8, 40, 44, 68, 81, 90, 108, 111, 116]). This is a concern as quinolones are antibiotics with high resistance potential and are listed on the WHO's Watch List of antibiotics that should be targeted for a reduction in use. Additionally, quinolones carry a higher risk for development of *Clostriodides difficile* colitis compared to other antibiotic classes [117]. In Australia, quinolone use is restricted in the community (and hence LTCFs) by the requirement that doctors obtain special authority at the time of prescribing. As a result of this national regulation, quinolone use in Australian LTCFs is low, despite overall antibiotic use in Australian LTCFs being high compared to other regions. A positive finding with respect to antibiotics on the WHO Watch List was that macrolides were reported to be among the three most frequently prescribed antibiotics in only four studies, one from France [74], one from the US [40], and two from Japan [67, 100].

Our findings indicate low rates of appropriate use of antibiotics in LTCFs, and, as assessed by the McGeer criteria, each year was associated with lower odds of antibiotic prescriptions being appropriate. However, this result is heavily based on estimates from Australia (six of eight studies) which show a decreasing trend in appropriateness between 2010–2018. Between 2006 and 2015 in Australia, the people entering LTCFs were older and more frail, a trend which likely contributes to increases in antibiotic use overall, as well as inappropriate use [30, 118]. However, the changes in the health and demographics of the LTCFs population would not account entirely for the decrease in antibiotic appropriateness, which requires further investigation. Furthermore, measurement of appropriateness across more countries and regions is required for a more robust estimation of regional and temporal trends.

The McGeer criteria were originally developed for infection surveillance in LTCFs [119] and to assess whether an infection is present, and thus whether treatment with an antibiotic is warranted. By way of comparison, seven studies included in our review used national or local guidelines to assess whether the type and dose of antibiotic used was appropriate (S8 File). Though we were not able to pool these results in a meta-analysis, levels of appropriate antibiotic choice were typically above 70%. This demonstrates how the choice of appropriateness algorithm or guideline affects estimates of appropriateness, and the need for studies to be clear in what features of antibiotic appropriateness they are assessing. Furthermore, based on the results of the included studies, the initiation of antibiotic treatment in LTCFs should be a key area for antimicrobial stewardship programs.

Our analysis did not find a change in antibiotic use prevalence in LTCFs over the study period (1985–2019 for point prevalence and 1985–2017 for 12-month period prevalence). This may be due to the small number of studies in earlier years (see S10 File). However, national antibiotic consumption levels are changing at varying rates between countries. For example, between 2000 and 2010, national antibiotic consumption was shown to be decreasing in the US, Canada, Spain, France, and Germany; but increasing in Australia, the United Kingdom, and several Asian countries [2]. In Australia, the first decreases in national antibiotic consumption were reported in 2015 and have persisted till now [120]. Thus, though we did not detect an overall trend in antibiotic use in LTCFs, this does not preclude changes in antibiotic use rates within regions and countries over time, which our analysis was not powered to detect. This raises the important issue of surveillance of antibiotic use in LTCFs by countries. Ideally, the data collected would allow the measurement of temporal trends. Point prevalence surveys, which are the main source of antibiotic use data in LTCFs, have limited ability to do this unless they are conducted for many years. Other data sources such as electronic databases of medication supply or electronic records from LTCFs provide the ability to monitor trends over time using multiple key indicators in addition to prevalence, such as DOT per 1000 resident days and number of courses per 1000 resident days [10, 30, 92, 116].

There are several limitations to our analysis. Firstly, we were not able to combine estimates of the DOT/1000 days, DDD/1000 days and number of courses/1000 days with meta-analysis due to differences in the reporting of these outcomes. These outcomes account for the length of treatment (DOT, DDD), the dose of treatment (DDD), and the frequency of treatment (number of courses). Thus, they are more sensitive to detecting changes over time than the prevalence of antibiotic use. Standardising the reporting of these outcomes in studies to facilitate the compilation of results is important for ongoing surveillance efforts. Secondly, there was heterogeneity in the antibiotic use rates reported in the meta-analyses. The variation in antibiotic use between facilities has been reported in previous studies and reviews [4, 6, 44, 50, 121]. A strength of our analysis was that we explored whether region and year of measurement explained this heterogeneity using meta-regression; however, we were unable to adjust for resident demographics and prevalence of key health conditions as planned due to inconsistencies in the way these were reported. We also ensured we were comparing systemic antibiotic use rates and rates of use in LTCFs for older adults. The meta-regression models for point prevalence and the prevalence of appropriate antibiotic courses explained 56% and 85% of the between study variance, respectively. Lastly, our review did not identify data from countries in the regions of Asia (aside from Singapore), South America or Africa. This is a key limitation of current surveillance efforts in LTCFs since antibiotic use is growing at rapid rates in middle-income countries [2].

Antibiotic use in LTCFs has gained substantial attention nationally and internationally in recent years as evidenced by the steady growth in the number of monitoring activities and intervention studies (see S10 File) [16, 20, 62]. However, growing concerns of inappropriate

antibiotic use are yet to be addressed by the widespread development of national antibiotic stewardship activities specific to LTCFs in many countries. The majority of antibiotic steward-ship in LTCFs to date have been local initiatives involving small numbers of facilities and deliv-ered mixed results [18, 20]. The European Centre for Disease Prevention and Control reported that of the 1043 LTCFs participating in the HALT-2 survey of 2013, 46% did not have any anti-biotic stewardship elements in place, and 76.4% did not have a list of antibiotics for restricted use [18]. There was no substantial improvement in the presence of these antibiotic stewardship elements in the 2016–17 HALT survey [62]. While many countries release national strategies to combat antimicrobial resistance that address human antibiotic use, these tend to focus on strategies and targets for community and hospital antibiotic use [122–124]. Our results high-light the need for specific strategies for LTCFs, particularly since LTCFs serve a unique popula-tion with complex care needs distinct from the general population.

## Conclusions

Concerted efforts are needed to tackle inappropriate use of antibiotics in LTCFs. In this review, we have summarized the body of literature on antibiotic use in LTCFs over a 35-year period (1985–2019) and compared antibiotic use among regions. We have highlighted key areas requiring action, including regions without data, regions with higher antibiotic use in LTCFs and the common use of antibiotics with high resistance potential, namely quinolones. Our analysis provides a regional and overall baseline against which to monitor progress in reducing antibiotic use in LTCFs.

## Supporting information

**S1 File. PROSPERO registration.**
(PDF)

**S2 File. PRISMA checklist.**
(DOCX)

**S3 File. Search strategy.**
(DOCX)

**S4 File. Quality assessment score for studies as assessed against Joanna Briggs Institute Critical Appraisal Tool for Prevalence Studies.**
(DOCX)

**S5 File. Point prevalence meta-analysis results, reported resident characteristics, and resi-dent eligibility.**
(DOCX)

**S6 File. Detailed heterogeneity statistics.**
(DOCX)

**S7 File. Resident characteristics and inclusion criteria reported in studies included in 12-month period prevalence antibiotic use meta-analysis.**
(DOCX)

**S8 File. Antibiotic use in long-term aged care facilities (studies not included in meta-analy-ses).**
(DOCX)

**S9 File. Resident characteristics reported in studies included in appropriateness of antibiotic use (according to McGeer criteria) meta-analysis.**
(DOCX)

**S10 File. Number of studies measuring antibiotic use in LTCF included in this systematic review by year.**
(DOCX)

## Acknowledgments

We would like to thank Mary Simons, Jeremy Cullis and Jane van Balen, Macquarie University Clinical Librarians for assistance with designing the search strategy for this review.

## Author Contributions

**Conceptualization:** Magdalena Z. Raban, Johanna I. Westbrook.

**Data curation:** Magdalena Z. Raban, Peter J. Gates, Claudia Gasparini.

**Formal analysis:** Magdalena Z. Raban.

**Funding acquisition:** Magdalena Z. Raban, Johanna I. Westbrook.

**Investigation:** Peter J. Gates.

**Methodology:** Magdalena Z. Raban, Peter J. Gates, Claudia Gasparini.

**Project administration:** Magdalena Z. Raban.

**Resources:** Johanna I. Westbrook.

**Supervision:** Magdalena Z. Raban.

**Writing – original draft:** Magdalena Z. Raban.

**Writing – review & editing:** Peter J. Gates, Claudia Gasparini, Johanna I. Westbrook.

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
