## [Decision Letter · Decision Letter 0]

21 Jul 2021

PONE-D-21-16655

Temporal and regional trends of antibiotic use in long-term aged care facilities across 39 countries, 1985-2019: systematic review and meta-analysis

PLOS ONE

Dear Dr. Raban,

Thank you for submitting your manuscript to PLOS ONE. After careful consideration, we feel that it has merit but does not fully meet PLOS ONE’s publication criteria as it currently stands. Therefore, we invite you to submit a revised version of the manuscript that addresses the points raised during the review process.

Reviewers have raised questions on the discrepancies and data gaps in the study 

We look forward to receiving your revised manuscript.

Kind regards,

Iddya Karunasagar

Academic Editor

PLOS ONE

3. We note that this manuscript is a systematic review or meta-analysis; our author guidelines therefore require that you use PRISMA guidance to help improve reporting quality of this type of study. Please upload copies of the completed PRISMA checklist as Supporting Information with a file name “PRISMA checklist”.

Additional Editor Comments (if provided):

Please see the reviewer comments. There seems to be some discrepancy in the stated years of study and the data included in the study. The authors also need to address the question on macrolide and sulphonamides. Please address al reviewer questions point by point.

Reviewers' comments:

Reviewer's Responses to Questions

**Comments to the Author**

1. Is the manuscript technically sound, and do the data support the conclusions?

Reviewer #1: Partly

Reviewer #2: Yes

2. Has the statistical analysis been performed appropriately and rigorously? 

Reviewer #1: Yes

Reviewer #2: Yes

3. Have the authors made all data underlying the findings in their manuscript fully available?

Reviewer #1: Yes

Reviewer #2: Yes

4. Is the manuscript presented in an intelligible fashion and written in standard English?

Reviewer #1: Yes

Reviewer #2: Yes

5. Review Comments to the Author

Reviewer #1: 1. The review seems more of Europe representation. The other areas have statistically less representation as one two studies from Singapore or 1 from Australia. This could bring about bias.

2. The said period was 1990 to 2021. Studies from 1985 are shown to be included.

3. Need to look into the data of Macrolides and sulphonamides use. These are widely used in the community setup for respiratory or urinary tract infections.

4. It would be interesting to note if the quinolones usage was before 2015 or it remains so even after watch list was announced by WHO.

5. Would be interesting to know if appropriate means only the drug or the duration and dosage was also considered. Most of the times the drug appropriatness will be met but the other criteria of stewardship will not be.

Reviewer #2: I must congratulate the authors for a very detailed analysis of an important topic in a neglected patient care setting i.e long term healthcare facilities. The statistical analysis tools are detailed and help bring clarity to the conclusions made.

6. PLOS authors have the option to publish the peer review history of their article (what does this mean?). If published, this will include your full peer review and any attached files.

Reviewer #1: No

Reviewer #2: No

---

## [Author Response · Author response to Decision Letter 0]

3 Aug 2021

Please see attached response to reviewers.

---

## [Decision Letter · Decision Letter 1]

9 Aug 2021

Temporal and regional trends of antibiotic use in long-term aged care facilities across 39 countries, 1985-2019: systematic review and meta-analysis

PONE-D-21-16655R1

Dear Dr. Raban,

We’re pleased to inform you that your manuscript has been judged scientifically suitable for publication and will be formally accepted for publication once it meets all outstanding technical requirements.

Kind regards,

Iddya Karunasagar

Academic Editor

PLOS ONE

Additional Editor Comments (optional):

All reviewer comments have been addressed satisfactorily

Reviewers' comments:

Reviewer's Responses to Questions

**Comments to the Author**

1. If the authors have adequately addressed your comments raised in a previous round of review and you feel that this manuscript is now acceptable for publication, you may indicate that here to bypass the “Comments to the Author” section, enter your conflict of interest statement in the “Confidential to Editor” section, and submit your "Accept" recommendation.

Reviewer #1: All comments have been addressed

2. Is the manuscript technically sound, and do the data support the conclusions?

Reviewer #1: Partly

3. Has the statistical analysis been performed appropriately and rigorously? 

Reviewer #1: Yes

4. Have the authors made all data underlying the findings in their manuscript fully available?

Reviewer #1: Yes

5. Is the manuscript presented in an intelligible fashion and written in standard English?

Reviewer #1: Yes

6. Review Comments to the Author

Reviewer #1: The comments raised in the previous review have been addressed well. But detailed description of the stewardship would be appreciated.

7. PLOS authors have the option to publish the peer review history of their article (what does this mean?). If published, this will include your full peer review and any attached files.

Reviewer #1: No

---

## [Editor Report · Acceptance letter]

13 Aug 2021

PONE-D-21-16655R1 

Temporal and regional trends of antibiotic use in long-term aged care facilities across 39 countries, 1985-2019: systematic review and meta-analysis 

Dear Dr. Raban:

I'm pleased to inform you that your manuscript has been deemed suitable for publication in PLOS ONE. Congratulations! Your manuscript is now with our production department. 

Kind regards, 

on behalf of

Dr. Iddya Karunasagar 

Academic Editor

PLOS ONE